# Quality-of-Life Assessment after Head and Neck Oncological Surgery for Advanced-Stage Tumours

**DOI:** 10.3390/jcm11164875

**Published:** 2022-08-19

**Authors:** Paula Luiza Bejenaru, Bogdan Popescu, Alina Lavinia Antoaneta Oancea, Catrinel Beatrice Simion-Antonie, Gloria Simona Berteșteanu, Mihnea Condeescu-Cojocarița, Anca Ionela Cîrstea, Irina Doinița Oașă, Teodora Elena Schipor-Diaconu, Dan Popescu, Raluca Grigore

**Affiliations:** 1Department 12-Otorhynolaryngology, Ophthalmology, Faculty of Medicine, “Carol Davila” University of Medicine and Pharmacy, 020021 Bucharest, Romania; 2Otorhinolaryngology Department, “Colțea” Clinical Hospital, 030171 Bucharest, Romania; 3Otorhinolaryngology Department, “Dr. N. Kretzulescu” Ambulatory Diagnostic and Treatment Medical Center, 050042 Bucharest, Romania; 4Otorhinolaryngology Department, “Dr. Carol Davila” Central Military Emergency University Hospital, 010825 Bucharest, Romania; 5Otorhinolaryngology Department, Emergency County Hospital, 130095 Targoviste, Romania

**Keywords:** patient-reported outcome measures (PROMs), quality of life, head and neck cancer

## Abstract

Squamous cell carcinoma of the head and neck (HNSCC) is a common malignancy often diagnosed in the advanced stage with a complex negative influence on the patient’s quality of life (QoL). Given its multi-modal treatment, the first step is to adequately balance the needs of the patient, and the second step includes the consultations, interventions, and care provided by the medical team, with the purpose of improving the overall management of the HNSCC. Current attempts to develop and validate quality-of-life instruments specific to cancers of the head and neck have been reported, and certain questionnaires are now available. We performed a retrospective study in a tertiary centre, involving 89 patients who survived 3 years after HNSCC surgery. A patient-related outcome measurement was made using the European Organization for Research and Treatment of Cancer (EORTC) QLQ-C30 and QLQ-H&N35 instruments to assess QoL at admission and 3 years after treatment. The 3-year survivors reported an overall improvement in QoL compared with those in the pre-treatment period. The unique details of head and neck cancer treatments outline the importance of considering the characteristics of the patient population in quality-of-life research and also identify how quality-of-life data can contribute to the care provided by the multi-disciplinary team involved in a patient’s follow-up.

## 1. Introduction

Head and neck squamous cell carcinoma (HNSCC) accounts for approximately 700,000 new cases each year worldwide, making it the eighth most common malignancy [1]. Most patients are diagnosed with a locally advanced disease requiring complex multi-modal treatments that involve surgery or sometimes multiple surgeries, either with or without reconstruction procedures; radiotherapy (RT); or chemo radiotherapy (CRT) [1,2]. Adequate synergy between the needs of patients and the consultations, interventions, and care provided by the medical team is essential for improving the overall management of HNSCC [3,4].

Quality of life is defined by the WHO as “individuals’ perceptions of their position in life in the context of the culture and value systems in which they live and in relation to their goals, expectations, standards and concerns” [5]. It is a broad-ranging concept incorporating in a complex way the person’s physical health, psychological state, level of independence, social relationships, personal beliefs, and relationships to salient features of the environment [5].

In recent years, the measurement of health has expanded beyond traditional health indicators, such as mortality and morbidity, to include measures of the impact of disease and impairment on daily activities and behaviour [6], perceived health measures [7], and disability/functional status measures [8]. These measures, whilst beginning to provide a measure of the impact of disease, do not assess quality of life per se, which has been aptly described as “the missing measurement in health” [9]. HRQoL (health-related quality of life) is more specific and allows medical professionals to appreciate the impact of disease and treatment on the patient psychologically, physically, and socially [5,9].

The EORTC quality-of-life questionnaire (QLQ) is an integrated system for assessing the health-related quality of life (HRQoL) of cancer patients participating in international clinical trials [9]. The core questionnaire QLQ-C30, which we used in our study (See Appendix A), is composed of both multi-item scales and single-item measures. These include five functional scales, three symptom scales, a global health status/QoL scale, and six single items. Each of the multi-item scales includes a different set of items—no item occurs in more than one scale. All of the scales and single-item measures range in score from 0 to 100. A high scale score represents a higher response level. Thus, a high score for a functional scale represents a high/healthy level of functioning, a high score for the global health status/QoL represents a high QoL, but a high score for a symptom scale/item represents a high level of symptomatology/problems. The scaling technique described above is based upon the widely applied Likert method of summated scales, in which the constituent items within each scale are simply summed. This makes several assumptions about the nature of the items, the most important of which are (a) that it is appropriate to give equal weight to each item and (b) that each item is graded on a linear or equal-interval scale. The raw QLQ-C30 scores can be transformed to scores ranging from 0 to 100 [10].

The rationale for using the EORTC QLQ was based on our experience with general population data based on large random samples from the general population in Norway, Sweden, and Denmark. The questionnaire is also accompanied by a Summary Booklet with examples of the use of statistical packages and the EORTC QLQ-C30 Reference Values Manual [10].

In summary, QoL scales can become reference tools for a longitudinal follow-up of head and neck cancer patient cohorts and can even inspire physicians to adapt and improve treatments according to patients’ socio-economical features.

### Aim

An assessment of QoL–general health and well-being, physical activity, role functioning, emotional functioning, cognitive functioning, social functioning, and symptom scales at a 3-year follow-up after surgical HNSSC treatment—and a comparison between the 3-year results and admission values.

## 2. Materials and Methods

We performed a retrospective cohort study on a group of 89 patients, all HNSCC survivors, at their 3-year follow-up appointments at a tertiary cancer centre hospital in Romania. All of the patients were diagnosed and treated in 2019 in the ENT Department of Coltea Clinical Hospital, Bucharest, Romania. All patients answered a Romanian-validated EORTC QLQ-30and QLQ-H&N35 at admission and another one at the 3-year follow-up (See Appendix A). All of them agreed to participate in the study and signed a consent form. The study was conducted according to the guidelines of the Declaration of Helsinki and approved by the Institutional Ethics Committee of Coltea Clinical Hospital (15 January 2019).

The patients were guided by a trained registered nurse and by a family member in completing the answers since we used a self-reported questionnaire, which was possible only in 60% of the cases. Due to age-related difficulties (poor sight and arthritis) or functional illiteracy, the remainder of the patients needed help in completing the answers. The inclusion criteria were an advanced stage of HNSCC (IVa) and survival at the 3-year follow-up; the exclusion criterion was any morbidity that could affect severely the patient’s participation (neurological or psychiatric diseases).

The admission files were evaluated, and descriptive statistics were performed using Microsoft Excel 2016, V16.0, Microsoft, Redmond, Washington, United States, and IBM^®^ SPSS® Statistics, V26.0, IBM, Armok, New York, NY, USA.

## 3. Results

Of 251 HNSCC patients diagnosed in an advanced stage (IVa) and treated in one year (2019) at Coltea Clinical Hospital, only 38% (N = 95) survived until the 3-year follow-up. Of those who did survive, 6 patients were lost at the 3-year follow-up. The group distribution was as follows: 41 patients with laryngeal first involvement (46.1%), 20 patients with hypopharyngeal first involvement (22.5%), 15 patients with oropharyngeal first involvement (16.9%), and 13 patients with oral cavity first involvement (14.6%) (Table 1).

The American Joint Committee on Cancer (AJCC)’s TNM classification system was used, and the results are summarized in Table 2 and Table 3.

The patients’ demographic characteristics were as follows: 78 patients were men (87.6%), and 11 were women (12.4%); the median age was 58 years; 72 patients were smokers (80%); 45 patients admitted to heavy alcohol consumption (50%); 29 patients had previous morbidities (32%)—cardiovascular diseases, diabetes mellitus, hepatitis, or pulmonary tuberculosis; 8 patients had previous RT (9%), and the rest of them had RT postoperatively; and 3 patients (3%) had CHT previously (all from the pharyngeal group). The sex distribution, TNM, and morbidities are shown below (Table 4 and Table 5).

All cases were discussed by the hospital’s tumour board, which guided the final treatment, and they all gave their informed consent for surgery. We do not have data regarding the HPV status of the entire group.

Good general health was reported by 80% of the patients, and 75% thought that their health was the same as or better than it was prior to the treatment.

At admission, the total laryngectomy group (larynx and hypopharyngeal sites of involvement), N = 61, reported the mean value of the following scales: QoL and general health and well-being, 41.7; physical activity, 60; role functioning, 66.7; emotional functioning, 41.7; cognitive functioning, 50; social functioning, 66.7 (Figure 1). On the symptom scale, they reported fatigability, pain, insomnia, dyspnoea, and financial difficulties.

A total of 84% (N = 51) of the patients suffered a total laryngectomy with primary vocal prosthesis (VP) placement through a tracheo-oesophageal fistula, and 16% (N = 10) of them were guided to learn to use their oesophageal voice or to use an electrolarynx.

At the 3-year follow-up, the VP group reported a mean value of 75 for the QOL and general health and well-being scale, 86.7 for physical activity, 66.7 for role functioning, and 100 for emotional, cognitive, and social functioning (Figure 1). They still reported an 11.1 on the scale for fatigability, 16.7 for pain, 33.3 for dyspnoea, 66.7 for insomnia, and 100 for digestive symptoms (constipation and/or diarrhea). As for symptoms, they reported sticky saliva and coughing (75%). In contrast, the patients without VP reported a mean value of 60 on the scale for QOL and general health and well-being, 60 for physical activity, 33.3 for role functioning, 50 for emotional functioning, 66 for cognitive functioning, and 33.3 for social functioning. They reported an 11.1 on the scale for fatigability and 66.7 for digestive symptoms (constipation and/or diarrhea), with a value of 50 for financial difficulties.

The oropharyngeal group, N = 15, reported at admission the following mean scales: QOL and general health and well-being, 25; physical activity, 46.7; role functioning, 33.3; emotional functioning, 25.0; cognitive and social functioning, 33.3. On the symptom scale, pain and insomnia were the most-reported symptoms. At the 3-year follow-up, patients reported the following mean scales: 41.7 for QOL and general health and well-being, 60.0 for physical activity, 33.3 for role functioning, 41.7 for emotional functioning, 50.0 for cognitive functioning, and 66.7 for social functioning (Figure 2). The symptom scales at the 3-year follow-up reported pain, dry mouth, and difficulties in opening the mouth.

The oral cavity group, N = 13, of which 10 required partial mouth-floor resection and some form of mandibulectomy, reported at admission the following mean scales: QOL and general health and well-being, 33.3; physical activity, 60.0; role functioning, 60; emotional functioning, 60; cognitive functioning, 75; and social functioning, 75. At the 3-year follow-up, they reported 41.0 for QOL and general health and well-being, 53.3 for physical activity, 33.3 for role functioning, 25.0 for emotional functioning, 33.3 for cognitive functioning, and 33.3 for social functioning (Figure 3). The symptom scales reported at the 3-year follow-up were pain, difficulties in opening the mouth, sticky saliva, and social contact.

## 4. Discussion

A quantitative QoL assessment can be performed by choosing from a large range of available tools. McHomey estimated in 2003 that almost 75 scales are used in oncology, and the choice of tool is fundamental to the study design and should be made according to the study’s objective, target population characteristics (head and neck cancer patients), and the psychometric properties of the scale [11,12]. There are two types of scales, generic and specific, and it is established that specific scales which address a certain disease/symptom/treatment are more sensitive to clinical variations and treatment effects than others are [11,12]. In addition, self-administered questionnaires are more sensitive. A validated scale meets the criteria of validity, reliability, and sensitivity. The validated scales specific to head and neck oncology, according to Heutte at al., are the QLQ-H&N35, a supplementary module to QLQ-C30, elaborated by the EORTC Group; Fact-H&N; University of Washington Questionnaire (UW-QOL); M. D. Anderson Symptom Inventory–Head & Neck (MDASI-HN); Head and Neck Performance Status Scale (PSS-HN); University of Michigan Head and Neck Specific Quality of Life Instrument and Head and Neck Cancer Inventory; Auckland Quality of Life Questionnaire (AQLQ). All these scales have either certain dimensions that are addressed or items specific to head and neck cancer treatments. The most frequently utilized and tested were: EORTC QLQ-H&N35, UWQOL, FACT-HN, and the University of Michigan Head and Neck Cancer Inventory [11]. Certain scales address just a symptom or treatment for a range of symptoms. For example, scales specific to voice and speech include the Vocal Handicap Index (VHIVHI-10), Speech Handicap Index, University of Michigan Voice-related Quality of life, Voice Prosthesis Questionnaire, Self-Evaluation of Communication Experiences after Laryngectomy, Voice Activity Participation Profile, Voice symptom scale, Voice performance, Parole Handicap Index. There are also scales specific to mucositis and xerostomia; swallowing and mastication; shoulder pathology; dental and feeding pathology; fatigue; oral pain; anxiety, depression, and psychological impact; sexuality, sleep, and alcohol consumption; cognitive features; independence; support [12].

Furthermore, finding the right instrument to assess a concept as large as the HRQoL is rather difficult, and some authors advocate that there cannot be a universal scale due to cultural and geopolitical differences. Even countries from the same continent can address pathologies with different angles, without deviating from the international guidelines, but with particularities due to their health system and socioeconomic development [13]. In addition, there is no gold standard questionnaire, which is due to the volume and heterogeneity of QoL measures [13]. Researchers should consider psychometric properties, research objectives, study design and pitfalls, and the benefits of combining different measures [11,13].

In our study, we used a validated Romanian EORTC QLQ-C30, version 3.0, developed by the European Organization for Research and Treatment of Cancer, Quality of Life Unit, Brussels, Belgium, in order to assess QoL at admission and at the 3-year follow-up (See Appendix A). This questionnaire was easy to complete, and patients declared in 89% of the cases that they thought this could be a tool to increase compliance. We found that the QLQ-H&N35 module in correlation with QLQ-C30 was, in most cases (56%), incorrectly completed, and patients declared that they found it tedious to answer so many questions.

The qualitative assessment is a new tool for physicians that allows them to see the disease in the context of the patient’s everyday life, with an open angle on physical data. The participant-observer experimental type is time-consuming, but it is a novel attitude in head and neck oncology, complementary to quantitative research [10].

In general, after HNSSC treatment, the overall quality-of-life (QoL) scores reported were around 70, and the operating scale scores were around 80; fatigue was the most common general symptom, while problems related to speech, sexuality, and oral function (salivary, teeth, and mouth-opening problems) were the most common symptoms of head and neck damage [14]. Some studies highlight factors such as tobacco use, psychological distress, and sex as the main determinants of long-term QoL. The correlation between tobacco use, psychological suffering, and QoL is complex and is mentioned by several authors [15,16]. As a result, interventions around smoking cessation and addressing psychological disorders could be direct actions for improving the system responsible for QoL. In addition, in the demographic factors of QoL impairment, some studies advocate that women are associated with lower indices [14]. The demographic characteristics of our group were similar to those found in the literature, with a higher prevalence of HNSSC in men with a history of smoking and alcohol consumption and a median age of 58. Of 251 patients diagnosed in an advanced stage (IVa) and treated in one year (2019), only 38% (N = 95) survived; the group’s heterogeneity was due to the high percentage of patients with total laryngectomy (69%, N = 61). A small percentage of patients had previous oncological treatments—9% RT and 3% CHT—and were all from the hypopharyngeal group, without significant influence among the QOL scores.

The main concern of a patient who survives an HNSCC is the fear of recurrence, followed by dental problems, salivation, fatigue, and speech and eating disorders. In terms of addressability, the patient usually focuses on addressing the coordinating surgeon of the case before addressing the speech therapist and the oral rehabilitation team [14]. Psychological distress is cited as the main determinant of long-term QoL, and the patient's age and gender significantly affect their needs and concerns [14].

The study of patients’ quality of life requires the application of standardized questionnaires, which are validated by the medical community, and the most commonly used questionnaires are UW in the United States and EORTC H&N in Europe [17,18]. The results of a QoL meta-analysis of survivors at one year after treatment concluded that generally the overall QoL of patients deteriorated from pre-treatment to 1–6 months thereafter but gradually improved thereafter to 12 months [17]. All studies analyzed reported a similar pattern during the examination period, although the change in the mean score between baseline and 12 months was not statistically significant (*p* > 0.05) [17]. Despite the general improvements in symptoms over time reported by many studies, xerostomia, changes in saliva consistency, and increased severity of fatigue influenced QoL in a statistically significant way at the one-year follow-up [17]. Other symptoms, such as difficulties in managing physical appearance, speech, taste/smell, sexuality, or swallowing, were also reported by at least one study to be significantly worse at 12 months [17]. Numerous studies showed that pain improved when compared with the pre-treatment period, and the anxiety score was also better than it was pre-treatment. Some studies showed that age was a determinant of QoL, but the data around the influence of sex on QoL are inconclusive. A multivariate analysis showed that the patient’s sex would have no influence on QoL [17]. Psychological distress, the ability to eat in society, and concern about physical appearance were significantly associated with impaired QoL at the one-year follow-up. Specifically, depressive symptoms at baseline could also significantly predict overall QoL at 12 months [17]. In our study, the QLQ scales seemed to provide better outcomes at the 3-year follow-up, but some of the symptoms, especially in the oral cavity and oropharyngeal groups, seemed to be worse at follow-up.

Regarding the QoL of patients with oral cancer, a longitudinal study that looked at 10-year-old survivors reported some improvement in physical appearance, chewing food, general condition, and anxiety but reported deterioration in swallowing. The study used the questionnaire developed by the University of Washington—University of Washington Quality of Life (UW-QOL) [18]. A 10-year survival of up to 34% was reported, with factors influencing age, tumour stage, reconstructive surgery with free flaps, and the primary form of treatment (*p* < 0.05). The affected results included appearance, mood, saliva, and shoulder function [19]. In our group, the QoL scales after oral cavity cancer treatment had mean values around the average or below, with 41.0 for QoL and general health and well-being. The mean values reported were lower than those from admission: 53.3 for physical activity, 33.3 for role functioning, 25.0 for emotional functioning, 33.3 for cognitive functioning, and 33.3 for social functioning. Conversely, the symptom scale results were higher, meaning the results were worse than the admission ones, with an increase in fatigue, pain, insomnia, digestive symptoms, and financial difficulties.

In 2020, Ranta et al. published an extensive study on the quality of life of oropharyngeal cancer survivors at an average follow-up time of 11.79 years [19]. Although the study model was a retrospective one, the results provide a topical overview of this type of cohort [19]. A retrospective analysis of diagrams and patient responses to the questionnaire on quality of life created by the European Organization for Cancer Research and Treatment, the core module (EORTC QLQ-C30), the head and neck module (EORTC QLQ-H&N35) and M.D. Anderson Dysphagia Inventory (MDADI) questionnaires represented the study bases [19]. Thus, they reported that, of the 263 survivors diagnosed and treated between 2000 and 2009, with a participation of 62.4% in the study (N = 164), most survivors reported a good QOL index [19]. The median overall health status of EORTC QLQ-C30 was 75.00 (IQR = 31.25). The one-way treatment group had significantly better quality-of-life outcomes than the combined treatment group. Non-smokers and previous smokers had significantly better quality-of-life outcomes than patients who smoked at the time of diagnosis. Additionally, heavy alcohol consumption was associated with worse results [19]. Patients with p16-positive cancer had significantly better QOL results than patients with p16-negative. Patients dependent on a percutaneous endoscopic gastrostomy tube (PEG) reported a significantly poorer quality of life than patients without a PEG tube [19]. The conclusion of this national study was that long-term QOL in OPSCC survivors is generally good. According to the previous literature, the one-way treatment was superior to the combined treatment in terms of long-term QOL results and should be followed whenever possible [19]. Similar data were reported from studies in Denmark, but there are very few studies with relevant results reported in developing countries. In our study, 3-year follow-ups reported mean values with better scores than the admission ones, though values were still around average—lower by 16.7 for QoL and general health and well-being, 13.3 points for physical activity, no change in points for role functioning, lower by 16.7 for emotional functioning, lower by 16.7 for cognitive functioning, and lower by 33.4 for social functioning. In the symptoms area, patients declared a lower fatigability; the same amount of pain; less dyspnoea, appetite loss, and constipation. However, they declared the same amount of financial difficulties. All patient survivors at the 3-year follow-up were patients with a transmandibular approach, without regional flap reconstruction. All reconstruction was made by local advancement flaps or with primary closure. Four patients had local complications: two had osteoradionecrosis, one rejected the titanium plate, and all of them developed fistulas.

The treatment of advanced laryngeal and hypopharyngeal cancer has undergone important changes in the sense that the transition from total laryngectomy (TL) to laryngeal preservation therapies is held due to the general perception that TL has a significant negative impact on a patient’s life. However, it has not yet been determined whether TL-related physical impairments translate into a health-related reduced quality of life (HRQoL). This paradigm shift began around the 1990s when the first of two large randomized clinical trials found that advanced laryngeal cancer could be treated with chemotherapy or radiation concomitant with an overall survival rate equivalent to that of TL with postoperative radiotherapy [20]. However, laryngeal cancer is the only cancer with a decrease in survival in the United States in recent decades, where the relative 5-year survival rate from 1975 to 1989 was 66%, compared with a 63% survival rate between 2005 and 2011 [19]. TL is a treatment that profoundly changes a patient’s quality of life, and people who have undergone such surgery are at risk of wound infection and necrosis, fistulas, dysphagia, voice problems, impaired breathing, and loss of smell [20]. A systematic review of the literature comparing QoL in patients with advanced laryngeal cancer treated with organ preservation versus similar patients treated with surgery indicated that there are not enough studies of sufficient quality to draw any conclusions on this topic.

Despite the overall improvement in voice rehabilitation, the presence of a permanent tracheostomy after total laryngectomy has a negative impact on a patient’s postoperative quality of life [20,21]. There has therefore been an increasing emphasis on laryngeal conservation in the development of new treatment strategies for patients with laryngeal cancer. Rather than focusing on the conservation of the larynx as an organ, the preservation of its function is the ultimate challenge in these procedures [21].

In our study, the VP group related a higher mean value for functional scales and a lower one in comparison with admission—QoL and general health and well-being with 33.3 points, physical activity with 26.7 points, no change in points for role functioning, emotional functioning with 58.3 points, and social functioning with 33.3 points. On the symptom scales, patients declared less fatigue (lower with 44.4 points), no nausea and vomiting, less pain (lower with 50 points), some degree of dyspnoea (33.3), the same degree of insomnia, a higher percentage of constipation, and the same degree of financial difficulties (Figure 1). Patients without VP declared a lower QoL and general well-being than those in the VP group but higher than their admission scores and further lower scores on functional scales than those in the VP group. The patients without VP declared mean values for the symptom scales lower than those in the VP group, with only 33 points for insomnia, 66 for constipation, and 50 for financial difficulties.

What may significantly change the QoL score is the currently very good quality vocal rehabilitation. A review of the literature which included studies encompassing 1085 laryngectomy patients undergoing voice rehabilitation, of which 869 (80.1%) were treated with voice prosthesis (VTE) while 216 (19.9%) were treated with oesophageal speech (EV), indicated that the cumulative VHI (Voice Handicap Index) results showed a significantly better score for the VTE (tracheoesophageal voice) group than for the EV (oesophageal voice) group (31.93 ± 12, 11 compared with 35.39 ± 20.6; *p* = 0.003). Since there was no significant difference recorded in VrQoL, the study claimed that VTE and EV are both effective procedures in voice rehabilitation after laryngectomy [21]. Although VTE allows for significantly better speech performance, it does not necessarily correlate to high VrQoL [21]. However, VTE is the current gold standard in postlaryngectomy vocal rehabilitation [21].

Our study has some limitations: A lack of a statistical correlation, a focus on descriptive analysis, and small groups. Further studies should take into consideration a timeline with a shorter follow-up period, which implies prospective studies. Additionally, multi-centre studies could give an adequate overview of QoL in developing countries.

## 5. Conclusions

QoL evaluation after HNSSC surgical treatment can be difficult to assess and compare, especially long-term, due to group and treatment heterogeneity and low survival rates. At 3-year follow-ups, compared with the other groups, our data report a better QLQ-C30 QoL score among the TL group, with slightly better outcomes in the VP group. Surgical treatment after oropharyngeal and oral SCC may imply lower QoL scores, even at a 3-year follow-up. 

HNSCC survivors may benefit from early screening for potential rehabilitation needs and require involvement in preventive rehabilitation programs prior to surgery when possible. Studies concluded that an improved swallowing function, decreased pain and discomfort, and reduced utilization of the feeding tube are predictive factors for QoL after HNSCC surgery, even long after the surgical treatment. Therefore, rehabilitation programs should address these aspects. QoL research in correlation with the characteristics of the patient population, framed by the unique details of head and neck cancer treatments, may provide a multi-disciplinary caregiver team with a better picture at follow-up.

## Figures and Tables

**Figure 1 jcm-11-04875-f001:**
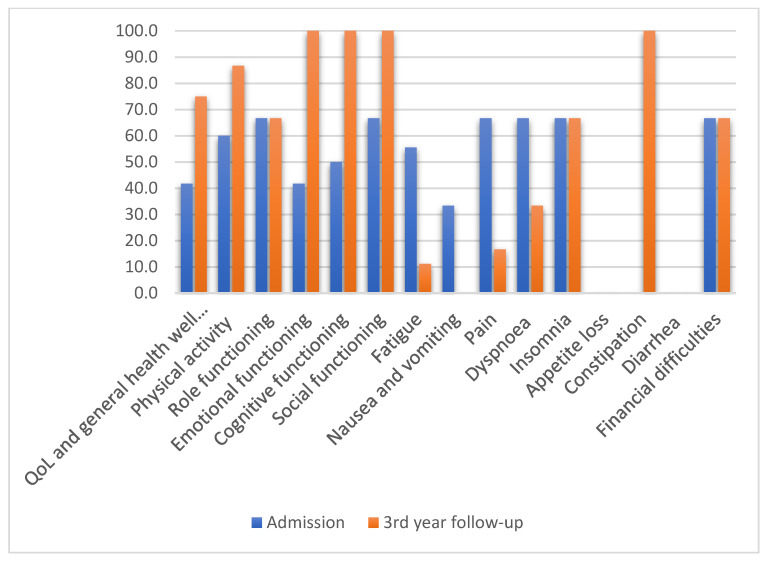
QLQ-C30 mean values after total laryngectomy with VP (N = 51).

**Figure 2 jcm-11-04875-f002:**
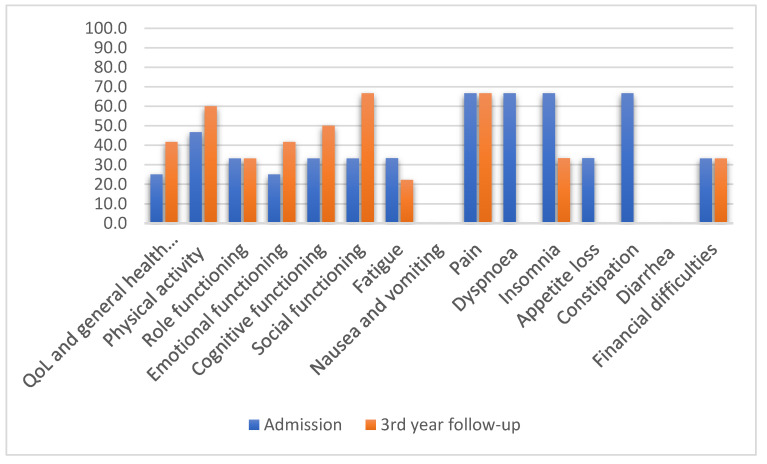
QLQ-C30 mean values after oropharyngeal cancer treatment (N = 15).

**Figure 3 jcm-11-04875-f003:**
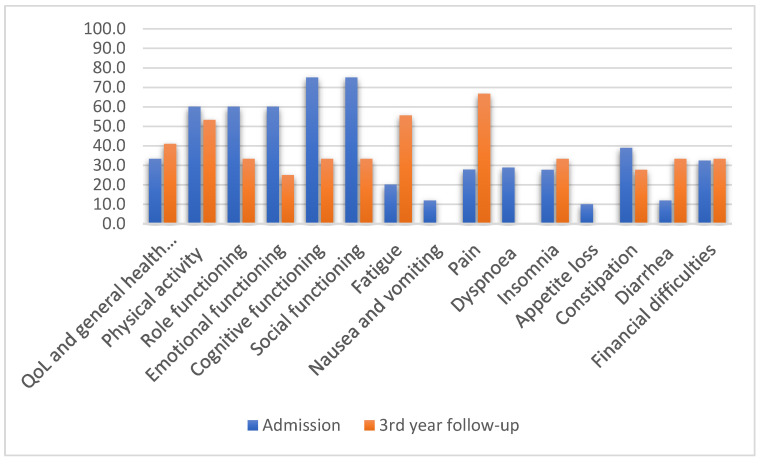
QLQ-C30 mean values after oral cavity cancer treatment (N = 13).

**Table 1 jcm-11-04875-t001:** Site of the primary tumour.

		Frequency	Percent
First site	Larynx	41	46.1
	Oral cavity	13	14.6
	Oropharynx	15	16.9
	Hypopharynx	20	22.5
	Total	89	100.0

**Table 2 jcm-11-04875-t002:** Site of the primary tumour T.

Count				
		T		Total
		3	4a	
First site	Larynx	1	40	41
	Oral cavity	4	9	13
	Oropharynx	9	6	15
	Pharynx	0	20	20
Total		14	75	89

**Table 3 jcm-11-04875-t003:** Site of the primary tumour N.

Count								
		N						Total
		0	1	2a	2b	2c	3	
First site	Larynx	1	13	15	11	1	0	41
	Oral cavity	0	0	0	6	7	0	13
	Oropharynx	0	0	1	5	8	1	15
	Pharynx	0	0	1	15	4	0	20
Total		1	13	17	37	20	1	89

**Table 4 jcm-11-04875-t004:** First site, sex distribution, and morbidities.

		With Morbidities	Total
First site of primary tumour	Larynx	21% (19)	41
	Oral cavity	2% (2)	13
	Oropharynx	6% (5)	15
	Hypopharynx	4% (3)	20
Total		33% (29)	89
Sex	F	2% (2)	11
	M	31% (27)	78

**Table 5 jcm-11-04875-t005:** Sex distribution of tumours T and N.

Count										
		T		N						Total
		3	4a	0	1	2a	2b	2c	3	
Sex	F	2	9	0	1	2	6	2	0	11
	M	12	66	1	12	15	31	18	1	78
Total		14	75	1	13	17	37	20	1	89

## Data Availability

Data are available from the authors at request.

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
