# Peer review of "Quality-of-Life Assessment after Head and Neck Oncological Surgery for Advanced-Stage Tumours"

_jcm, 2022, doi:10.3390/jcm11164875_

Round 1
Reviewer 1 Report
The authors made an effort in writing this paper.
Abstract: there are 2 aims of this study (as stated in abstract)- literature review and retrospective study. Later in the abstract, there are no data related to literature review. In addition, there are no information about second aim- the authors should me more clear about this retrospective study- they should state which variables were investigated. The lines 22-25 are not necessary for the abstract section.
Introduction- is too long, i suggest putting some of the paragraphs (i. e lines63-97) in discussion section, from the abstract it seems that literature review is put in this section. Also, the authors should state why was specific questionnary was used.
Aim: is not included after the introduction.
Material and method: the authors should add their questionnaire (in Romanian and English) as supplementary file.
Results: should be presented also as tables or figures.
Discussion: The authors should compare their study with others, not only to describe the results of the others.
Conclusion- can be improved. Some of the conclusions are not directly related to the results of their study. Some of the sentences are only the contemplations of the authors.
Author Response
Dear Reviewer,
Thank you very much for the time you taken to revise this manuscript. Your comments are very important and useful for us in order to improve our work. You can find the clarifications in the attached document.
[Abstract: "there are 2 aims of this study (as stated in abstract)- literature review and retrospective study. Later in the abstract, there are no data related to literature review. In addition, there are no information about second aim- the authors should me more clear about this retrospective study- they should state which variables were investigated . The lines 22-25 are not necessary for the abstract section." -> a shorter abstract, with a single aim stated, lines 22-25 deleted.
Introduction- "is too long, i suggest putting some of the paragraphs (i. e lines63-97) in discussion section, from the abstract it seems that literature review is put in this section. Also, the authors should state why was specific questionnary was used." -> lines 63-97 became lines 224-261; lines 122-16
Aim: " is not included after the introduction." lines 130-134
Material and method: "the authors should add their questionnaire (in Romanian and English) as supplementary file. " - we will add it
Results: "should be presented also as tables or figures." - lines 161- 222
Discussion: "The authors should compare their study with others, not only to describe the results of the others." - lines 262-267; 284-291; 335-338;346-352;376-386;411-422
Conclusion-" can be improved. Some of the conclusions are not directly related to the results of their study. Some of the sentences are only the contemplations of the authors." -> lines 437-445]
I would like to thank you once again, in advance, for your hard work.
Paula Bejenaru

Reviewer 2 Report
Thank you for let me reviewing the presente work.
The Authors give a complete and clear up-to date introduction and literature review about QOL in HN patients and descrive the questionario available in literature.
Unfortunately I have some suggestions ti point out:
1) It would bè interesting ti have also a short term follow up of the patients with a QOL questionarie performer short term After Surgery.Some of the results infact are not expected such as the improved global quality of Life or social score After surgery.
2) The results are Better show in tables or graphs, i suggest to summarize the results in tables.
3) The Authors only list symptoms before and after surgery without reporting the Number or percentage of patients that complain these symptoms.
4) Data of the reconstructive Surgery in the Oral cavity Surgery (Free flaps, no reconstruction, pedicled flaps) have to be reported, such as the voice restoration in totale laryngectomy with tracheoesophageal puncture and prothesis or not. The rehabilitation and reconstruction data are of vital importance becose affect the results of the QoL.
5) Why are oropharingeal neoplasie treated with Surgery without assessing HPV positivity in your Center?
6) Some spelling errors have to be improved
Author Response
Dear Reviewer,
Thank you very much for the time you taken to revise this manuscript. Your comments are very important and useful for us in order to improve our work. You can find the clarifications in the attached document.
1) It would bè interesting ti have also a short term follow up of the patients with a QOL questionaire performer short term After Surgery. Some of the results in fact are not expected such as the improved global quality of Life or social score After surgery. -> I am grateful for this observation, we also started a prospective study with this ideea. On this study, we focused on the patients who came at 3rd year follow-up and compared the results with the admission ones. It is true that in the first year after surgery, QoL is strongly influenced by the treatment, as it is also stated in the literature, but we were curious if in our region, results improve in time.
2) The results are Better show in tables or graphs, i suggest to summarize the results in tables. -> lines 161- 222
3) The Authors only list symptoms before and after surgery without reporting the Number or percentage of patients that complain these symptoms. -> lines 161- 222
4) Data of the reconstructive Surgery in the Oral cavity Surgery (Free flaps, no reconstruction, pedicled flaps) have to be reported, such as the voice restoration in totale laryngectomy with tracheoesophageal puncture and prothesis or not. The rehabilitation and reconstruction data are of vital importance becose affect the results of the QoL. -> lines 376-386, lines 411-422
5) Why are oropharingeal neoplasie treated with Surgery without assessing HPV positivity in your Center? lines 176-178
6) Some spelling errors have to be improved
I would like to thank you once again in advance for your hard work.
Paula Bejenaru

Round 2
Reviewer 2 Report
Thank you for revising your manuscript. The Authors improved the whole manuscript, and now it is in my opinion suitable for beeing published.
Author Response
Dear Reviewer,
Thank you for the effort and expertise that you contributed towards reviewing the article. We really appreciate all your comments and suggestions and we are also grateful for your last appreciation.
Yours sincerely,
Paula Bejenaru.